# Relationships between Flexural and Bonding Properties, Marginal Adaptation, and Polymerization Shrinkage in Flowable Composite Restorations for Dental Application

**DOI:** 10.3390/polym13162613

**Published:** 2021-08-06

**Authors:** Akimasa Tsujimoto, Masao Irie, Erica Cappelletto Nogueira Teixeira, Carlos Alberto Jurado, Yukinori Maruo, Goro Nishigawa, Takuya Matsumoto, Franklin Garcia-Godoy

**Affiliations:** 1Department of Operative Dentistry, University of Iowa College of Dentistry, 801 Newton Rd., Iowa City, IA 52242, USA; erica-teixeira@uiowa.edu; 2Department of Biomaterials, Okayama University Graduate School of Medicine, Dentistry and Pharmaceutical Science, 2-5-1, Shikata-cho, Kita-ku, Okayama 700-8525, Japan; mirie@md.okayama-u.ac.jp (M.I.); tmatsu@md.okayama-u.ac.jp (T.M.); 3Woody L. Hunt School of Dental Medicine, Texas Tech University Health Sciences Center El Paso, 5001 El Paso Drive, El Paso, TX 79905, USA; carlos.jurado@ttuhsc.edu; 4Department of Occulusion and Removable Prothodontics, Okayama University, 2-5-1, Shikata-cho, Kita-ku, Okayama 700-8558, Japan; ykmar@md.okayama-u.ac.jp (Y.M.); goro@md.okayama-u.ac.jp (G.N.); 5Department of Biosciene Reserach, University of Tennessee Health Science Center, 875 Union Ave., Memphis, TN 38163, USA; fgarciagodoy@gmail.com

**Keywords:** flexural strength, bond strength, marginal adaptation, polymerization shrinkage, resin composite

## Abstract

To evaluate the flexural and bonding properties, marginal adaptation, and polymerization shrinkage in flowable composite restorations and their relationships, four new generation flowable composites, one conventional, and one bulk-fill flowable composite were used in this study. Flexural properties of the composites and shear bond strength to enamel and dentin for flowable restorations were measured immediately and 24 h after polymerization. Marginal adaptation, polymerization shrinkage, and stress were also investigated immediately after polymerization. The flexural properties, and bond strength of the flowable composites to enamel and dentin were much lower immediately after polymerization than at 24 h, regardless of the type of the composite. Polymerization shrinkage and stress varied depending on the material, and bulk-fill flowable composite showed much lower values than the others. The marginal adaptation and polymerization shrinkage of the composites appeared to have a much stronger correlation with a shear bond strength to dentin than to enamel. The weak mechanical properties and bond strengths of flowable composites in the early stage after polymerization must be taken into account when using them in the clinic. In addition, clinicians should be aware that polymerization shrinkage of flowable composites can still lead to the formation of gaps and failure of adaptation to the cavity regardless of the type of composite.

## 1. Introduction

The use of dental amalgams in restorative dentistry has been reduced since the Minamata convention in 2013, which demanded a global effort for the gradual elimination of mercury-containing materials from dentistry, given its potential release in the oral environment [1]. Japan and Scandinavian countries have either prohibited amalgam or restricted its use over the past decade. Other countries, such as Canada and Germany, recommend keeping amalgam restorations out of the mouths of children, pregnant people, and people with kidney impairment. Because of this, direct composites restorations have become increasingly popular for larger cavities in posterior teeth [2].

Available resin composites for direct restorations can be classified into two types based on differences in viscosity: (1) packable [3]; and (2) flowable composites [4]. Flowable composites were first introduced in the late 1990s, and their mechanical properties have improved over the past 20 years [4], expanding their clinical applications to include large cavities in posterior teeth [5]. An investigation of the clinical usage of flowable composite in posterior teeth in Germany from February to April of 2010 reported that 78.6% of the clinicians preferred to use flowable composites for posterior restorations and in larger cavities to apply as a cavity liner [6]. However, a systematic review published in 2016 concluded that the application of flowable composite as a cavity liner for posterior restorations did not reduce microleakage or improve clinical performance [7]. In recent years, however, flowable composites with reduced polymerization shrinkage/stress and improved mechanical properties have been brought to market [8,9]. Tsujimoto et al. also showed that the polymerization shrinkage/stress of new generation flowable composites was further reduced by horizontal or oblique layering techniques [10]. In addition, clinical studies of flowable composite restorations in posterior teeth using layering techniques reported similar performances to those of packable composites over a period of two to three years [5,11,12].

Recently, the popularity of the bulk-fill technique for large cavities in posterior teeth has been increasing due to the development of bulk-fill composites [13]. However, in the American Dental Association clinical evaluators panel report for posterior composite restorations, 70% of the respondents preferred composites used with a layering technique, and only 26% of them tended to use composites with the bulk-fill technique [14]. This seems to be due to concerns about the inadequate depth of cure and relatively high polymerization shrinkage/stress when using the bulk-fill technique with bulk-fill composites. As a result, the use of flowable composites with layering techniques in larger cavities of posterior teeth has been increasing [5,11,12].

Nevertheless, the polymerization shrinkage and stress, and mechanical properties of flowable composites are still concerns for clinicians dealing with large posterior restorations. The polymerization shrinkage and stress of flowable composites may lead to the creation of gaps if adhesion to cavity floor and walls is not adequate, which in turn can lead to microleakage and secondary caries [15]. In addition, composite restorations in posterior teeth are subjected to a wide range of external forces such as food mastication and unconscious bruxism [16]. If the forces applied to flowable composite restorations exceed the strength of the material itself, fracture or marginal gaps may occur [17]. Therefore, it is important to investigate these materials through many different aspects, such as the mechanical and bonding properties, marginal adaptation, and polymerization shrinkage of new generation flowable composites.

In addition, the correlations between polymerization shrinkage stress and mechanical properties, especially elastic modulus, of composites have been evaluated, but the evidence for the correlation of these factors with composites is currently conflicting and may indicate changing relationships. Gonçalves et al. in 2010 showed strong relationships (*R*^2^ = 0.966) between polymerization shrinkage stress and elastic modulus [18]. However, in Bicalho et al. in 2014, polymerization shrinkage stress showed much weaker correlations (*R* = 0.567, *p* = 0.111 for the strain gauge method) to the elastic modulus of materials [19]. In reports from 2020, the correlation between polymerization shrinkage stress and the elastic modulus of composites was measured as *R* = 0.22, *p* = 0.761 [8]. This transition could be potentially attributed in part to the improvement of flowable composites and in part to concerns that the evaluations did not appropriately reflect polymerization shrinkage reactions because those studies evaluated the elastic modulus using flexural strength measurement 24 h after polymerization. As polymerization shrinkage stress occurs especially in the initial stages after polymerization, the most relevant mechanical properties are those of the composite immediately after polymerization [20]. However, there is no reported research on these values, and thus in this study, the correlations between mechanical and bonding properties immediately after polymerization and polymerization shrinkage-related parameters were evaluated. The first null hypothesis to be tested was that the properties of new generation flowable composites will not differ from those of conventional or bulk-fill flowable composites. In addition, a second null hypothesis was that none of these values would be correlated.

## 2. Materials and Methods

The 4 new generation flowable composites used in this study were: (1) Beautifil Flow Plus X F03 (BF, Shofu, Kyoto, Japan), (2) Clearfil Majesty ES Flow Low (CM, Kuraray Noritake Dental, Tokyo, Japan), (3) Estelite Universal Flow Medium Flow (EU, Tokuyama Dental, Tokyo, Japan), and (4) G-ænial Universal Injectable (GU, GC, Tokyo, Japan). A single conventional flowable composite developed 15 years ago, Unifil LoFlow Plus (UP, GC), and 1 bulk-fill flowable resin composite, Filtek Bulk Fill Flowable (FF, 3M Oral Care, St. Paul, MN, USA), were used for comparison. The flowable composites used in this study are indicated in Table 1.

In this study, each flowable composite was used with the adhesive recommended by the manufacturer. The 4 light-cure universal adhesives used in this study were: (1) Beautifil Bond Universal (Shofu) for BF, (2) Clearfil Universal Bond (Kuraray Noritake Dental) for CM, (3) G-Premio Bond (GC) for GU and UP, and (4) Scotchbond Universal Adhesive (3M Oral Care) for FF. A single chemical-cure universal adhesive was used: Tokuyama Universal Bond (Tokuyama Dental) for EU. The universal adhesive used in this study and manufactures instructions are indicated in Table 2 and Table 3.

### 2.1. Measurement of Flexural Properties

Specimen preparation and measurements of flexural properties were conducted in accordance with ISO 4049 with a Teflon split mold (25 × 2 × 2 mm), which was developed by Irie et al. [21] (Figure 1), rather than in a stainless-steel mold. When attempting to measure flexural properties at the initial stage after polymerization, a Teflon split mold can minimize the stresses exerted on the specimen during removal from the mold. The composite was placed into the mold, the upper surface closed with a clear matrix strip (Epitex, GC), and the material pressed with a glass plate under a 5 N load. The exit window of a light-emitting diode (LED) dental curing light (Elipar Deep Cure-S LED curing light, 3M Oral Care) was placed against the glass plate and the sample light-cured with a total radiant power of 352,800 mJ/cm^2^ (1470 mW/cm^2^ × 40 s × 3 overlapping sections × 2 sides). Then, specimens were carefully removed from the mold and polished. For each condition for each material, 10 specimens were prepared.

Flexural properties were measured at two different times: 10 min after light-curing (IM), and after 24 h in distilled water storage at 37 °C (24 h). A 3-point bending test with a 20 mm span and a load speed of 0.5 mm/min was performed with a universal testing machine (Type 5565, Instron, Norwood, MA, USA). The flexural strength in MPa and elastic modulus in GPa were automatically determined from the stress-strain curve of the custom software package supplied by the manufacturer of the testing machine (Series IX software, Instron).

### 2.2. Shear Bond Strength

De-identified extracted human premolars, and third molars were used for shear bond strength and microleakage tests. A total of 360 human molars, extracted for orthodontic reasons, were used for the shear bond strength test. A low-speed precision cutter (IsoMet 1000, Buehler, Chicago, IL, USA) with a diamond saw was used to section the roots at 1 mm below the cement-enamel junction before the coronal portion of the tooth was sectioned into buccal and lingual halves. All pulp tissue was removed. Distilled water was used to ultrasonically clean the prepared teeth for 30 s, and they were then air-dried. The teeth were mounted in slow-setting epoxy resin (Epofix, Struers, Copenhagen, Denmark), which was placed under tap water to absorb heat from the polymerization reactions and limit any temperature rise. A grinder polisher was used to grind the surfaces of the coronal central portion with 120-, 400-, and 600-grit silicon carbide (SiC) papers (Sankyo Fuji Star, Saitama, Japan) under running tap water. Enamel and dentin surfaces with a standardized surface texture and smear layer were prepared using this method. The prepared surfaces were treated with the universal adhesive specified by the manufacturer and light-cured following the manufacturer’s instructions. A Teflon mold insert (3.6 mm diameter and 2.0 mm height) was positioned on the treated surface of the enamel or dentin and filled in one increment up to 2.0 mm height with the composite. The exit of the LED light curing unit (3M Oral Care) was fixed at the top surface of the mold insert, and the material was light-cured with a total radiant power of 58,800 mJ/cm^2^ (1470 mW/cm^2^ × 40 s). The curing tip of the unit was kept as close as possible to the surface during irradiation and vertically directly above it. The mold insert was then removed, and the finished specimens were divided into IM and 24 h groups. For each condition for each material, 10 specimens were prepared.

In the IM group, the shear bond strength test was conducted 10 min after polymerization using the universal testing machine at a crosshead speed of 1.0 mm/min. In the 24 h group, the specimens were stored in distilled water at 37 °C for 24 h, and then shear bond strength testing was carried out in the same way. The shear bond strengths (MPa) were calculated by dividing the peak load at failure by the bonding area (10.17 mm^2^).

### 2.3. Marginal Adaptation

A total of 60 human molars, extracted for orthodontic reasons, were used for the marginal adaptation test. Each human molar was embedded in the slow-setting epoxy resin. A flat enamel surface (4 mm diameter) was exposed using the grinder polisher with wet 180-grit SiC paper. With the tooth held rigidly in a custom-made drill press, a cylindrical cavity (3.5 mm diameter, 1.5 mm in depth) was prepared using both a tungsten carbide bur (200,000 rpm) and a custom bur (4000 rpm) under wet conditions. One cavity was prepared in each tooth in the coronal region and on the mesial surface. The cavity walls and surrounding enamel margin, without bevel, were treated with universal adhesive according to the manufacturers’ instructions. Each cavity was slightly overfilled with the composite (*n* = 10 per group), covered with a plastic strip, and cured. Excess filling material and approximately 0.1 mm of enamel were removed by wet grinding with 1200-grit SiC paper using a grinder-polisher, followed by polishing with linen using an aqueous slurry of Alfa Micropolish (0.3 μm; Buehler) immediately after polymerization. The presence, location, and extent of marginal gaps were determined using a traveling microscope (400×; XY-B, D-Type, Nikon, Tokyo, Japan). The maximum gap-width and the opposing width between the material and the cavity wall were measured 10 min after polymerization. The sum of these 2 measurements was defined as the marginal gap in the tooth cavity.

### 2.4. Polymerization Shrinkage

Specimen preparation and measurements of polymerization shrinkage were conducted by the bonded-disk method [22]. Composite specimens were placed in brass rings (15 mm diameter) fixed on a glass slab (3.0 mm thick). The interior of the brass ring was circular, while the material of the rings themselves had a square cross-section. Airborne particle abrasion with 50 μm alumina powder for 10 s was used to prepare the upper surface of the glass slab for bonding to the composite. The air pressure was set to 0.2 MPa, and the orifice of an airborne-particle abrader (Jet Blast II, J. Morita, Osaka, Japan) was held approximately 1 mm from the surface. The composite specimens (1.0 mm in height, 8 mm in diameter) were prepared by compressing composite (0.09 g) using a glass plate to obtain disks of the right dimensions. These disks were positioned centrally within the brass rings (10 mm internal diameter), leaving a free space around the composite disk. A flexible glass coverslip (22 × 22 × 0.1 mm) was placed thus that it was supported by the brass ring and in contact with the composite sample.

The bonded disk arrangement was secured on a custom jig. The jig was made of an aluminum stand with a horizontal stage for specimen placement, fitted with 2 stainless steel clips to hold the glass slab. The stage included a brass ring with a hollow center through which the tip of a light-curing unit was fixed in place. A clamp was attached to the stage to hold a uni-axial linear variable displacement transducer (LVDT) measuring system and allow for vertical adjustment of its position. The LVDT measuring system was positioned centrally over the coverslip. The specimens were light-cured with total radiant power of 32,000 mJ/cm^2^ (1470 mW/cm^2^ × 40 s) using the LED curing light (3M Oral Care). For each material, 6 specimens were prepared.

The signal from the LVDT was passed through a signal conditioning unit and a high-resolution analog to digital converter and data logger to be recorded in a computer. Following mechanical equilibration, data were captured every second for 60 min from 20 s prior to commencement of irradiation.

Six specimens were made and tested for each material. For each composite, the maximum shrinkage strain measured during each of the 3 runs was recorded, and the polymerization shrinkage in % was calculated.

### 2.5. Polymerization Shrinkage Stress

Specimen preparation and measurements of polymerization shrinkage stress were conducted by the aluminum cuspal deflection method [10]. Aluminum blocks (10 × 8 × 15 mm) were fabricated with a MOD cavity (4 × 8 × 4 mm) using a computer-aided design/ computer-aided manufacturing system, creating 2 different cusps. The cavity interior was airborne particle abraded with 50 μm alumina powder for 10 s to improve adhesion. The air pressure was set to 0.2 MPa, and the orifice was held approximately 10 mm from the metal surface when using the abrader. The universal adhesive specified by the manufacturer was applied prior to placing the composite according to the manufacturer’s instructions. The adhesives, except for Tokuyama Universal Bond, which is a chemical-cured adhesive, were light-cured with a total radiant power of 147,000 mJ/cm^2^ (1470 mW/cm^2^ × 10 s) at a standardized distance of 1 mm using an LED curing light. The composite was incrementally or bulk filled and light-cured for each layer or bulk with a total radiant power of 117,600 mJ/cm^2^ for incremental or 58,800 mJ/cm^2^ for bulk-filling (1470 mW/cm^2^ × 40 s × 2 increments with incremental filling technique or 1470 mW/cm^2^ × 40 s in bulk-filled). For each material, 6 specimens were prepared.

Two LVDT measuring systems were set up on 2 XYZ tables with 3 attached micrometers (Mitutoyo, Tokyo, Japan). The cuspal deflection detected by the LVDT measuring system was collected using data acquisition and analysis software. The sensitivity of the LVDT probes exceeded 0.1 μm in the range of ± 1 mm. Measurements of cuspal deflection were obtained over 600 s. The cuspal displacements measured at both cusps were added to produce the total deflection.

### 2.6. Statistical Analysis

Statistical analysis was conducted with a commercial statistical software package (SPSS Statistics Base, International Business Machines, Armonk, NY, USA). Because the Kolmogorov–Smirnov test confirmed the normal distribution of data, a 2-way analysis of variance (ANOVA) for flexural properties and shear bond strengths and 1-way ANOVA for marginal adaptation, polymerization shrinkage, and shrinkage stress, with Tukey’s post-hoc honest significant difference test (significance level of 0.05), were used for data analysis. Correlations among different indicator values were analyzed by linear regression (significance level of 0.05, adjusted by Bonferroni correction to 0.001).

## 3. Results

### 3.1. Flexural Properties

The results for the flexural properties of the tested flowable composites are shown in Table 4. The flexural strengths of new generation flowable composites were 51.3–102.4 MPa immediately after polymerization and 128.6–162.0 MPa after 24 h. Rank order immediately after polymerization was EU-BF-GU-UP-CM-FF, but this changed to GU-EU-CM-FF-BF-UP at 24 h due to the difference in compositions. The new generation flowable composites showed significantly higher values immediately after polymerization than those of the conventional (51.7 MPa) and bulk-fill (50.3 MPa) flowable composites. In addition, the flexural strengths of all new generation composites after 24 h were significantly higher than that of the conventional composite (92.9 MPa), and most of them were similar to that of the bulk-fill composite (144.9 MPa).

The elastic modulus of new generation flowable composites was 2.07–5.04 GPa immediately after polymerization and 7.51–9.24 GPa after 24 h. As for the flexural strength results, most of the new generation composites immediately after polymerization showed significantly higher values than those of conventional (1.80 GPa) and bulk-fill (1.44 GPa) composites. In addition, the elastic modulus of all new generation composites after 24 h was significantly higher than that of conventional (4.15 GPa) and bulk-fill (6.01 GPa) composites.

### 3.2. Shear Bond Strength

The results for the shear bond strength to dentin and enamel of the tested flowable composite restorations are shown in Table 3. The shear bond strengths of the new generation flowable composites in the self-etch mode were 14.0–15.2 MPa to enamel and 14.1–16.0 MPa to dentin immediately after polymerization, and 19.7–21.2 MPa to enamel and 19.1–21.4 MPa to dentin after 24 h. The bond strengths were similar to the conventional (17.8 MPa in immediate and 19.6 MPa in 24 h to enamel; 16.3 MPa in immediate and 20.2 MPa in 24 h to dentin) and bulk-fill (15.0 MPa in immediate and 21.7 MPa in 24 h to enamel; 16.3 MPa in immediate and 23.6 MPa in 24 h to dentin) composites regardless of storage conditions, and there were no statistically significant differences. In addition, the enamel and dentin bond strengths of recent and bulk-fill composites increased significantly over time, but the bond strengths of conventional flowable composites did not change regardless of storage conditions.

### 3.3. Marginal Adaptation

The results for the marginal adaptation of the tested flowable composite restorations are shown in Table 3. New generation flowable composites showed 57–79 µm for sum and 0.16–0.22% for the rate of marginal gap formation, results similar to the conventional (50 µm and 0.14%) and bulk-fill (62 µm and 0.17%) composites. There were no differences in marginal adaptation between the materials and material types.

### 3.4. Polymerization Shrinkage

The results for the polymerization shrinkage of the tested flowable composites immediately after polymerization are shown in Table 3. The polymerization shrinkage rate of new generation flowable composites was 3.18–3.95%, and significantly higher than that of bulk-fill composite (2.81%). The polymerization shrinkage rate of conventional was 3.48%, which was higher than EU and FB, similar to CM and GU, and lower than BF.

### 3.5. Polymerization Shrinkage Stress

The results for the polymerization shrinkage stress of the tested flowable composites immediately after polymerization are shown in Table 3. Changes in cusp distance in the aluminum block before and after polymerization of new generation flowable composites were 12.07–15.77 µm, and these values were significantly higher than those of conventional (10.13 µm) and bulk-fill (7.49 µm) composites.

### 3.6. Linear Regression Analysis

The results of linear regression analysis of the different indicator values are shown in Table 5. As there were only six flowable resin composites to compare, the statistical significance of these results was not high, and they should be approached with caution. Even so, the strong positive correlation (*R* = 0.9828) between flexural strength and elastic modulus measured immediately after polymerization was significant, even after applying the Bonferroni correction to allow for the fact that 55 correlations were calculated.

The strong positive correlation (*R* = 0.8344) between the marginal gap and immediate flexural strength was notable because there was no sign of a correlation between the marginal gap and flexural strength at 24 h (*R* = 0.0408). There was also a strong negative correlation (*R* = –0.9284) between the marginal gap and immediate shear bond strength to dentin. Many other correlations appear strong but must also be handled with caution.

## 4. Discussion

One important aspect of this study is that it investigated different time points after the polymerization of flowable composites. It is clear from the results of this experiment that the flexural properties and enamel and dentin bond strength of the flowable and bulk-fill flowable composites are much lower immediately after polymerization than after 24 h, regardless of the type of flowable composite. This is not a surprising result, but it does suggest that restorations, even with new generation flowable composites, are vulnerable to external forces applied at that point for morphological correction and occlusion adjustment, especially as they are also subjected to the internally generated polymerization shrinkage and stress. This suggests that clinicians should avoid, ideally and as far as possible, shaping the restoration immediately after polymerization, as this will increase the risks of damaging the restoration and bonding interface. That is, the composite should be placed as carefully as possible, correctly reproducing the morphology of the tooth, to avoid the need for a modification immediately after polymerization. The aim should be to minimize the finalization work that must be carried out at that point, even with the latest composites.

This is particularly true of the new generation flowable composites, where there is a much greater increase in mechanical properties over the first 24 h. The properties of the conventional flowable composite tested 24 h after polymerization was 92.9 MPa for flexural strength and 4.15 GPa for elastic modulus, while new-generation flowable and bulk-fill composites showed much higher values (128.6–62.0 MPa for flexural strength, 6.01–9.24 GPa for elastic modulus). This suggests, again, that clinicians should avoid, as much as possible, stressing the restorations immediately after polymerization,

These results also indicate that the mechanical and bonding properties of the composites in the initial stages after polymerization are important factors in the selection of flowable composites because the properties were different depending on the material. Further investigations of these properties are desirable and should be made use of in the development of new materials. The results show that even new generation flowable composites are still weak in the early stages, leaving plenty of room for improvement. Therefore, clinicians might instruct, at least, patients to be cautious when eating certain types of foods during the first 24 h after restoration placement.

On the other hand, the results for shear bond strength, both immediate and 24 h, and marginal adaptation showed no significant differences between materials or substrates. This means that the bonding performances of flowable composite restorations might not differ if the system recommended by the manufacturer is used. Thus, although it is important to bear the flexural properties results in mind when selecting a material, one can assume that the bond strength will be adequate throughout the process, independent of the system.

Here, the measured correlations are of interest. First, it is worth noting that polymerization shrinkage of the composites appeared to have a much stronger correlation with a shear bond strength to dentin than to enamel. The correlation between polymerization shrinkage and shear bond strength to dentin (*R* = 0.663, *p* = 0.1485 in the IM group, *R* = 0.7801, *p* = 0.0672 in 24 h group) was much higher than that with bond strength to enamel (*R* = 0.1202, *p* = 0.8206 in immediate group, *R* = 0.0973, *p* = 0.8545 in 24 h group). Although the statistical significance here is too low to draw any firm conclusions, this difference does suggest that further research may be valuable.

Similar observations can naturally be made about the correlation between shear bond strength and marginal adaptation. A recent study using real-time imaging to investigate the internal adaptation of composite restorations using swept-source optical coherence tomography showed that high brightness, which indicates microgaps or non-adapted areas, was observed at the cavity bottom on the dentin surface in all tested flowable composites using layering techniques immediately after light-curing [23]. That is, microgaps or non-adapted areas arise at the bottom of the cavity even in recent flowable composites, which the manufacturer claimed were designed to reduce polymerization shrinkage. The same research group also reported that only dual-cure flowable composite showed no internal gaps, unlike all the tested bulk-fill flowable composites, although the gaps were different in degree depending on the material [15]. In addition, the latest study from the group showed that a flowable composite, which was newly developed and designed for 3 s high irradiance light-curing did not show defect formation [24]. These studies suggest that it is difficult to achieve perfect adaptation at the bottom of a 4 mm deep cavity using most of the flowable composites, including bulk-fill type, which is used in the clinic. However, dual-cure or newly designed flowable composites performed better than earlier materials.

From the results of this study, it appears that the flexural properties of new generation flowable and bulk-fill composites have improved regardless of time since cure, and the technology for reducing polymerization shrinkage and stress has changed due to the development of bulk-fill composites, as was seen in the results. If we consider what the results of this experiment suggest for the improvement of the sealing of resin composites to dentin, we can say that further reductions in polymerization shrinkage and increases in initial dentin bond strength are important.

The implications for mechanical properties are less clear. It seems that it is not necessarily the case that these should be high from the initial stages, but rather start low and improve over time. To be specific, a strong negative correlation (*R* = −0.9284, *p* = 0.0075) was observed between immediate bond strength to dentin and the presence of marginal gaps. This is easy to understand, as the stronger the bond to the dentin, the less likely the resin is to pull away from it during polymerization. On the other hand, there were also strong correlations between marginal gap and flexural strength (*R* = 0.8344, *p* = 0.0389) and elastic modulus (*R* = 0.7861, *p* = 0.0637) immediately after polymerization; these correlations were not seen in the 24 h group (*R* = 0.0408 for flexural strength, *R* = 0.6165 for elastic modulus). This suggests that increasing the flexural strength and elastic modulus of composites immediately after polymerization may have negative side effects, and attention must be paid to ways to mitigate these effects if the mechanical properties of the composites are improved. On the other hand, improving the bonding performance to dentin would seem to be positive in all respects, at least as far as can be judged from these data.

In addition, significant correlations between polymerization shrinkage stress and elastic modulus immediately after polymerization (*R* = 0.8382, *p* = 0.0371) were seen, in contrast to the 24 h results (*R* = 0.6820, *p* = 0.1639). Generally, it has been considered that there are correlations in values between polymerization shrinkage stress and elastic modulus after 24 h [25]. However, this correlation should be re-evaluated as a correlation in values between the polymerization shrinkage stress and elastic modulus values at the initial stage of polymerization.

Overall, the first null hypothesis was rejected based on the results of this study. The properties of new generation flowable composites were clearly different from those of conventional and bulk-fill flowable composites. The second null hypothesis was not rejected, as the only statistically significant correlation found was between immediate flexural strength and immediate elastic modulus. However, the results strongly suggest that there was a correlation between the immediate mechanical properties and marginal gap, at least, and further work on a wider range of materials would be valuable.

## 5. Conclusions

The mechanical and bonding properties of new generation flowable composites are much lower immediately after polymerization than after 24 h. Polymerization shrinkage can lead to the formation of gaps and failure of adaptation to the cavity. This work suggests that the initial flexural strength, elastic modulus, and polymerization shrinkage, as well as the initial bond strength to dentin, may have a substantial influence on this behavior, and thus further research on flowable composite restorations at the initial stage after polymerization is important.

## Figures and Tables

**Figure 1 polymers-13-02613-f001:**
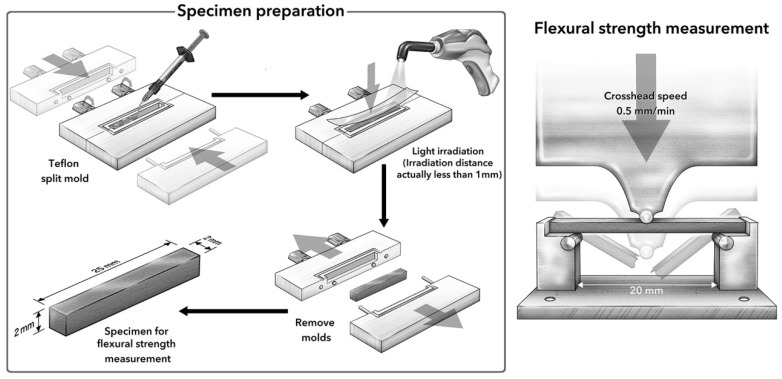
Schematic illustration of set-up for flexural strength measurement.

**Table 1 polymers-13-02613-t001:** Flowable composites used in this study.

Material (Abbrevistion)	Type of Composite (vol% of Fillers)	Main Components	Manufacturer
Beautifil Flow Plus X F03 (BF)	New generation flowable (50–60 vol%)	Bis-GMA, Bis-MPEPP, TEGDMA, Aluminofluoro-borosilicte glass filler, Photoinitiators, Accelerators, Pigments	Shofu, Kyoto, Japan
Clearfil Majesty ES Flow Low (CM)	New generation flowable (64 vol%, 78 wt%)	Hydrophobic aromatic dimethacrylate, TEGDMA, Silanated barium glass filler, Silanated silica filler, Photoinitiators, Accelerators, Pigments	Kuraray Noritake Dental, Tokyo, Japan
Estelite Universal Flow Medium Low (EU)	New generation flowable (57% vol%, 71 wt%)	Bis-GMA, Bis-MPEPP, TEGDMA, UDMA, Zirconia/silica filler, Photoinitiators, Accelerators, Pigments	Tokuyama Dental, Tokyo, Japan
G-ænial Universal Flo (GU)	New generation flowable (50 vol%, 69 wt%)	Bis-MPEPP, TEGDMA, UDMA, Silica filler, Strontium glass filler, Photoinitiators, Accelerators, Pigments	GC, Tokyo, Japan
Unifil LoFlow Plus (UP)	Conventional (42 vol%, 63 wt%)	UDMA, TEGDMA, Aluminofluoro-borosilicate glass filler, Photoinitiator, Accelerators, Pigments	GC, Tokyo, Japan
Filtek Bulk Fill Flowable Restorative (FF)	Bulk-fill (43 vol%, 65 wt%)	Bis-GMA, UDMA, Silica filler, Zirconia filler, Zirconia/silica cluster filler, Accelerators, Photoinitiators	3M Oral Care, St. Paul, MN, USA

Abbreviations: Bis-GMA, bisphenol A diglycidyl methacrylate; Bis-MPEPP, 2,2-Bis(4-methacryloxypolyethoxyphenyl)propane; TEGDMA, triethylene glycol dimethacrylate; UDMA, urethane dimethacrylate.

**Table 2 polymers-13-02613-t002:** Universal adhesives used in this study.

Universal Adhesive	Type of Polymerization	Main Components	Manufacturer
BeautiBond Universal	Light-cure	Bis-GMA, Carboxylic acid monomer, Phosphonic acid monomer, TEGDMA, Acetone, Water, Photoinitiators, Accelerator	Shofu, Kyoto, Japan
Clearfil Universal Bond	Light-cure	Bis-GMA, HEMA, Hydrophilic amide monomer, 10-MDP, Filler, Ethanol, Water, NaF, Photoinitiators, Accelerator, Silane coupling agent	Kuraray Noritake Dental, Tokyo, Japan
G-Premio Bond	Light-cure	MEPS, Methacrylate monomer, 4-MET, 10-MDP, Silica filler, Acetone, Water, Photoinitiators, Accelerator	GC, Tokyo, Japan
Scotchbond Universal Adhesive	Light-cure	Bis-GMA, HEMA, Decamethylene dimethacrylate, Ethyl methacrylate, Propenoic acid, Methyl-reaction products with decanediol and phosphorous oxide, Copolymer of acrylic and itaconic acid, Dimethylaminobenzoate, Methyl ethyl ketone, Silica filler, Ethanol, Water, Photoinitiators, Accelerator	3M Oral Care, St. Paul, MN, USA
Tokuyama Universal Bond	Chemical-cure	Liquid A: Bis-GMA, HEMA, MTU-6 Phosphoric acid monomer, TEGDMA, Acetone, Others Liquid B: γ-MPTES, Acryl borate catalyst, Peroxide, Acetone, Isopropyl alcohol, Water, Others	Tokuyama Dental, Tokyo, Japan

Abbreviations: Bis-GMA, bisphenol A diglycidyl methacrylate; HEMA, 2-Hydroxyethylmethacrylate; MTU-6, 6-methacryloyloxyhexyl-2-thiouracil-5-carboxylate; TEGDMA, triethylene glycol dimethacrylate; 4-MET, 4-methacryloxyethyl trimellitic acid; 10-MDP, 10-Methacryloyloxydecyl dihydrogen phosphate; γ-MPTES, γ-methacryloyloxypropyltriethoxysilane.

**Table 3 polymers-13-02613-t003:** Application Protocols for Tested Universal Adhesives.

Universal Adhesive	Application Protocol
BeautiBond Universal	Adhesive was applied to the air-dried surface for 10 s and then strong air pressure was applied over the liquid adhesive for 5 s or until adhesive no longer moved and the solvent was completely evaporated. Light-cured for 10 s.
Clearfil Universal Bond	Adhesive was applied to the air-dried surface for 10 s and then medium air pressure was applied over the liquid adhesive for 5 s or until adhesive no longer moved and the solvent was completely evaporated. Light-cured for 10 s.
G-Premio Bond	Adhesive was applied to the air-dried surface for 10 s and the strong air pressure was applied over the liquid adhesive for 5 s or until the adhesive no longer moved and the solvent was completely evaporated. Light-cure for 10 s.
Scotchbond Universal Adhesive	Adhesive was applied to the air-dried surface with a rubbing motion for 20 s and then medium air pressure was applied to the surface for 5 s. Adhesive was light cured for 10 s.
Tokuyama Universal Bond	Adhesive was applied to the air-dried surface for 10 s and then medium air pressure was applied over the liquid adhesive for 5 s. No light irradiation.

**Table 4 polymers-13-02613-t004:** Flexural Strength (FS), Elastic Modulus (EM), Shear Bond Strength to Enamel (SBS-E) and Dentin (SBS-D), Marginal Adaptation (MA), Polymerization Shrinkage (PS), and Polymerization Shrinkage Stress (PSS)of Flowable Composites immediately after polymerization (IM) and 24 h (Same superscript letters indicates no significant difference, *p* > 0.05).

Material (Type of Composite)	FS (IM) (MPa)	FS (24 h) (MPa)	EM (IM) (GPa)	EM (24 h) (GPa)	SBS-E (IM) (MPa)	SBS-E (24 h) (MPa)	SBS-D (IM) (MPa)	SBS-D (24 h) (MPa)	MA (%)	PS (%)	PSS (µm)
BF (New generation)	95.2 (9.5) ^a,A^	128.6 (6.0) ^a,B^	4.12 (0.47) ^a,A^	8.27 (0.71) ^a,B^	14.0 (1.7) ^a,A^	21.2 (3.0) ^a,B^	13.2 (3.4) ^a,A^	20.1 (4.5) ^a,B^	0.22 (0.03) ^a^	3.95 (0.16) ^a^	15.44 (0.72) ^a^
CM (New generation)	51.3 (4.3) ^b,A^	151.7 (5.4) ^b,B^	2.07 (0.17) ^b,A^	7.52 (0.28) ^a,B^	14.8 (2.6) ^a,A^	21.0 (3.1) ^a,B^	16.0 (2.2) ^a,A^	21.1 (4.0) ^a,B^	0.16 (0.07) ^a^	3.53 (0.08) ^b^	13.82 (0.56) ^b^
EU (New generation)	102.4 (10.6) ^a,A^	152.3 (8.5) ^b,c,B^	5.04 (0.60) ^c,A^	8.28 (0.49) ^a,B^	14.8 (3.2) ^a,A^	19.7 (3.4) ^a,B^	15.2 (3.4) ^a,A^	21.4 (3.6) ^a,B^	0.19 (0.08) ^a^	3.18 (0.07) ^c^	15.77 (0.74) ^a^
GU (New generation)	80.3 (4.1) ^c,A^	162.0 (9.1) ^c,B^	3.22 (0.33) ^e,A^	9.24 (0.59) ^c,B^	15.2 (2.9) ^a,A^	20.5 (3.7) ^a,B^	14.1 (2.9) ^a,A^	19.1 (2.9) ^a,B^	0.20 (0.08) ^a^	3.47 (0.10) ^b^	12.07 (0.34) ^c^
UP (Conventional)	51.7 (4.0) ^b,A^	92.9 (6.7) ^d,B^	1.80 (0.30) ^b,A^	4.15 (0.41) ^e,B^	17.8 (3.6) ^a,A^	19.6 (5.3) ^a,A^	16.3 (3.1) ^a,A^	20.2 (3.3) ^a,B^	0.17 (0.04) ^a^	3.48 (0.08) ^b^	10.13 (0.56) ^d^
FF (Bulk-fill)	50.3 (1.8) ^b,A^	144.9 (5.3) ^b,B^	1.44 (0.22) ^d,A^	6.01 (0.49) ^b,B^	15.0 (2.7) ^a,A^	21.7 (3.8) ^a,B^	16.3 (2.6) ^a,A^	23.6 (3.0) ^a,B^	0.14 (0.09) ^a^	2.81 (0.12) ^d^	7.49 (0.36) ^e^

Two-way analysis of variance (ANOVA) for flexural properties and SBS for immediate and 24 h groups, and one-way ANOVA for marginal adaptation, polymerization shrinkage/stress, with Tukey’s post-hoc honest significant difference test. Values in parentheses are standard deviations.

**Table 5 polymers-13-02613-t005:** Regression Analysis between Flexural Strength (FS), Elastic Modulus (EM), Shear Bond Strengths to Enamel (SBS-E) and Dentin (SBS-D), Marginal Adaptation (MA), and Polymerization Shrinkage (PS) and Stress (PSS) immediately after polymerization (IM) and 24 h.

Measured Indicators	*R* and *p* Values	FS (IM)	FS (24 h)	EM (IM)	EM (24 h)	SBS-E (IM)	SBS-E (24 h)	SBS-D (IM)	SBS-D (24 h)	MA	PS
FS (24 h)	*R*	0.3010	-	-	-	-	-	-	-	-	-
	*p*	0.5621									
EM (IM)	*R*	0.9828	0.3065	-	-	-	-	-	-	-	-
	*p*	0.0004	0.5547								
EM (24 h)	*R*	0.7026	0.8072	0.6846	-	-	-	-	-	-	-
	*p*	0.1195	0.0522	0.1335							
SBS-E (IM)	*R*	−0.5143	−0.7267	−0.4907	−0.7867	-	-	-	-	-	-
	*p*	0.2965	0.1019	0.3230	0.0634						
SBS-E (24 h)	*R*	−0.2599	0.3526	−0.3441	0.1699	−0.6082	-	-	-	-	-
	*p*	0.6189	0.4930	0.5042	0.7476	0.2002					
SBS-D (IM)	*R*	−0.7717	−0.2397	−0.6829	−0.7351	0.5627	−0.1215	-	-	-	-
	*p*	0.0723	0.6474	0.1349	0.0959	0.2450	0.8186				
SBS-D (24 h)	*R*	−0.3625	0.1257	0.3467	0.3484	−0.1670	0.4605	0.5978	-	-	-
	*p*	0.4801	0.8124	0.5008	0.4985	0.7518	0.3581	0.2101			
MA	*R*	0.8344	0.0408	0.7861	0.6165	−0.3275	−0.2376	−0.9284	−0.7499	-	-
	*p*	0.0389	0.9389	0.0637	0.1924	0.5263	0.6504	0.0075	0.0860		
PS	*R*	0.3331	−0.2628	0.326	0.2800	−0.1202	−0.0973	−0.663	−0.7801	0.7491	-
	*p*	0.5188	0.6148	0.5284	0.591	0.8206	0.8545	0.1485	0.0672	0.0865	
PSS	*R*	0.7637	0.2463	0.8382	0.6820	−0.4989	−0.2757	−0.6018	−0.4632	0.7265	0.6192
	*p*	0.0772	0.6380	0.0371	0.1639	0.2489	0.5969	0.2063	0.3549	0.102	0.1899

## Data Availability

Data are contained within the article.

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
