# Peer review of "Relationships between Flexural and Bonding Properties, Marginal Adaptation, and Polymerization Shrinkage in Flowable Composite Restorations for Dental Application"

_polymers, 2021, doi:10.3390/polym13162613_

Round 1

Reviewer 1 Report

General

This is an interesting topic of a well-designed and conducted study on the several short-term properties of flowable dental resin composites. I consider the lack of the same curing unit and different total radiant power values for all tests as a significant study flaw. Other minor issues are listed below:

Abstract

- Some ambiguous sentences need clarification: “Bulk-fill flowable composite showed much lower values than the other flowable groups. “ – which values? I presume pol. shrinkage and stress, but please make it clear for the readers.

- Also, please remove or rephrase the last sentence mentioning other studies. The abstract is no place for the results of other authors.

Introduction

-  The statement: “Available resin composites for direct restorations can be classified into two types based on viscosity and handling properties: (1) nanofilled [3]; and (2) flowable composites [4].” is quite bold as none of these references deal with classifications of contemporary resin composites, nor this is a classification based on viscosity (nanofilled?). Please elaborate or remove.

- There seem to be two null hypotheses in the manuscript: the first one at line 96 and the second one at line 124. Please enumerate them in the Introduction so that the readers can easily distinguish which one you are referring to.

The term “old-type” is an unfortunate phrase.- Different naming of the experimental groups seems necessary. I recommend “new generation flowable / (flowable) bulk fill / conventional flowable” or something else.

Materials and Methods

- Table 1 – please add a column with the vol%/wt% of fillers

- It would also be helpful to add material abbreviations to Table 1

- Flexural strength – the specimens were light-cured with 3 irradiations? Only one side? Please specify.

- Please state the number of specimens for Flexural strength.

- Nominal radiant exitance of a curing unit often declines over time, and it is necessary to measure it before use. What was the radiant exitance of your curing unit, and in which mode was it used?

- Please add another table with the manufacturers’ instructions for use for each adhesive. This is important for comparison purposes for future studies.

- Marginal adaptation – please specify whether the cavity margins were beveled and the curing parameters

- line 298, please specify that Bondmer Lightless is chemically cured.

Results

- Table 3, material Filtek is misspelled (Filteck)

- I commend your efforts to give a clear overview of all the results in one table. However, the differences among materials would be more apparent in diagrams.

- Please mention that there was no statistical difference between groups in marginal adaptation.

- Please try to fit Table 4 onto one page

Discussion

- You comment on the correlation between polymerization shrinkage stress and elastic modulus in the introduction. Still, there is no mention in the Discussion of the correlation of the immediate flexural modulus and polymerization shrinkage stress in your study, R= 0.8382, P=0.0371. Please discuss.

- Are there any other similar studies investigating the immediate and short-term properties of flowable composites? Please compare and comment.

Author Response

Thank you very much for your comments to improve the paper. I have attached the response to reviewer 1, so please see attachment.

Reviewer 2 Report

  • I think the title is a bit hanging. My suggestion is "Relationships between flexural and bonding properties, marginal adaptation, and polymerization shrinkage in flowable composite restorations for dental applications" ?
  • I suggest authors to summarize the abstract comprehensively. It is too long and some sentence is not necessary to mention, which I suggest not more than lin number 25.
  • Introduction: Good and satisfactory.
  • 2. Materials and Methods: Clear mentioned.
  • 2.1, This is interesting and novel research findings. Authors mentioned about flexural test for dental specimen (25×2×2 mm) together with he exit 157window of a light emitting diode (LED) dental curing light. For me in mechanical test for composite structure I never seen testing setup like that. I suggest authors to put Figure that elaborate on the testing setup and how it works.
  • Same goes to to other testing setup. It is novel for authors to share for at least with the schematic diagram on how it is as setup.
  • 3. Result and discussion: Flexural: Please ammend the way your presnt the tened of data. e.g 51.3-102.4 MPa into 51.3 MPa to 102.4 MPa. It is more significant.
  • the highest flexural strengh is Estelite Universal Flow Medium Low, but why ?. Then, the highest flexural modulus goes to Gracefil LoFlow, Why ? Authors did not mention any of that in term of scientific reasing ratehr than just a simple trend.
  • Same goes to other findings. What make each of the brand different? why ? Need to explain on each of the performance. Yes, I have read on the discussion parts,  I really hope that authorswill move their related explanations on discussions part to around each of the results area. I am sure this will drive readers to understand easily and not confuse.
  • Do you have any image to support  "much lower immediately after polymerization than after 24 h" to show the appearance of the sample ?
  • Conclusions need to improve comprehensively according to authors novelty findings. Also what is the resolutions based on 5 brand of flowable, which one is the good one ?

Author Response

Thank you very much for your comments to improve the paper. I have attached response to reviewer 2, so please find attachment.

Round 2

Reviewer 2 Report

Authors had done comprehensively corrections. I believe this manuscript will give a novel reference for future researchers and dental industry.